# Validating a Simple Mechanistic Model That Describes Weather Impact on Pasture Growth

**DOI:** 10.3390/plants10091754

**Published:** 2021-08-24

**Authors:** Edward B. Rayburn

**Affiliations:** Extension Service, West Virginia University, Morgantown, WV 26506-6108, USA; erayburn@wvu.edu

**Keywords:** pasture, growth, rotational, continuous, stocking, Appalachia, grazing, Python

## Abstract

Mathematical models have many uses. When input data is limited, simple models are required. This occurs in pasture agriculture when managers typically only have access to temperature and rainfall values. A simple pasture growth model was developed that requires only latitude and daily maximum and minimum temperature and rainfall. The accuracy of the model was validated using ten site-years of measured pasture growth at a site under continuous stocking where management controlled the height of grazing (HOG) and a site under rotational stocking at a West Virginia University farm (WVU). Relative forage growth, expressed as a fraction of maximum growth observed at the sites, was reasonably accurate. At the HOG site constant bias in relative growth was not different from zero. There was a year effect due to the weather station used for predicting growth at HOG being 1.8 km from the pasture. However, the error was only about 10-percent. At the WVU site constant bias for relative growth was not different from zero and year effect was eliminated when adjusted for nitrogen status of the treatments. This simple model described relative pasture growth within 10-percent of average for a given site, environment, and management using only daily weather inputs that are readily available. Using predictions of climate change impact on temperature and rainfall frequency and intensity this model can be used to predict the impact on pasture growth.

## 1. Introduction

Mathematical models allow researchers to package and evaluate knowledge and determine where information is missing. Sophisticated grassland ecosystem models are excellent for organizing and evaluating ecological relationships [1,2,3,4,5]. These models can be used at the local and national level for evaluating weather risk and the impacts of climate change on grassland-based livestock production [4,6,7]. However, sophisticated models require sophisticated input.

Models can also be used by practitioners to estimate the consequences of production inputs or environmental variables. Crop production models are useful at the local and national scale for estimating crop yields [6,7]. Pasture-based livestock producers seldom have detailed weather information available. Therefore there was a need for a pasture growth model that uses only basic weather station data for calculating the effect of weather on pasture growth. This is possible since weather impact on pasture growth has been studied extensively [8,9,10]. A mathematical model was developed that uses location latitude, daily minimum and maximum temperature, and daily precipitation to predict pasture growth. The output of this model was tested against pasture growth measured at two locations in the Alleghany Plateau of West Virginia in the USA.

## 2. Results

Measured cumulative forage growth observed (cumFGobs) was highly related to modelled cumulative relative growth rate due to environment (cum_rgr_env) (Figure 1a and Figure 2a, Table 1, A and E). Relative forage growth observed (relFGobs) had a standard deviation about the regression (SDreg) of 10% or less (Table 1, B and F). The relFGobs at HOG was directly related to cum_rgr_env with the regression intercept not being significantly different from zero (Table 1, B). At the WVU site there was a slight (8%) fixed bias in the observed versus modelled relation (Table 1, F). There were year effects on cumFGobs and relFGobs not accounted for by the pasture growth model at both sites. When year was included in the regression R^2^ increased and SDreg and AAPE decreased (Table 1, C, D, and G).

At the HOG site years broke into two groups. Years 1992, 1993, 1994 and 1999 differed in intercept but had similar slopes (Table 1, C). Years 1995, 1996, and 1997 showed little year effect (Table 1, D) with 1997 being slightly greater than the other two years. The year effect at this site is likely due to the weather station being on a hill 1.8 km away from and above the pasture. On spring nights hill tops have warmer nighttime temperatures than the valley below due to cool night air draining into the valley. On clear windless nights these valleys are “frost pockets” with temperatures cool enough to allow frost to form on vegetation in the low-lying areas while there is no frost on the hill tops. This will impact the cumulative temperature effecting initiation of plant growth and temperature effective on plant growth in the spring. Likewise, summer rainfall occurring as convection storms will provide water on one area but miss another area as far away as 1.8 km. Thus rainfall occurring at the weather station may not have occurred on the pasture and visa-versa.

At WVU there was a year by poultry litter interaction not described by the model (Table 1, G). For pastures not receiving poultry litter there was no year effect (Table 1, H). For pastures receiving poultry litter there was a year effect (Table 1, I and J). In the first year (1997), there was low growth response (Table 1, I) compared to the following two years (1998 and 1999) (Table 1, J). This is explained by the fact that nitrogen in poultry litter is slowly available. As poultry litter decomposes about one-half of the nitrogen is available the first year and the remaining nitrogen becomes available over the following years [11] (p. 215). In the first year, the pasture growth rates on the litter treatments were no better than on the no-litter treatments. The third study year (1999) had the highest relFGobs vs cum_rgr_env as a response to nitrogen. In that year, the plants had nitrogen available from litter applied in that year and the residual N from the previous two years of applied litter. After accounting for the differences in nitrogen availability over years there was little to no unaccounted-for year effect at the WVU site.

## 3. Discussion

This simple pasture growth model is reasonably descriptive of observed pasture growth. At the HOG site the year effect error was relatively small ranging 10 percent above and below the mean (Figure 2b and Table 1, C). The two years having the greatest effect (1993 and 1999) were major drought years with low summer rainfall which the model accounted for. At the WVU site, pastures not receiving poultry litter had no year effect (Table 1, H). Pastures receiving litter had a year effect due to nitrogen availability over time. After adjusting for nitrogen availability there was no major year effect (Table 1, I and J).

The pasture growth modelled rgr_env was quite accurate in describing the effect of drought. In the drought year 1999 the end of year rgr_env was 92 and FGobs was 3323. In 1998, a non-drought year, end of year rgr_env was 174 and FGobs was 5736. Using 1998 values as the base, the drought of 1999 reduced forage growth and rgr_env by 58 and 53%, respectively (3323/5736 = 0.58, 92/174 = 0.53) compared to 1998. Thus the pasture growth model estimate of drought impact on forage growth was within 10% of that measured in the field (53/58 = 0.91).

This model can be expanded for use with warm-season forages by changing the relation of evapotranspiration to pan evaporation to 0.75 and adjusting the minimum plant growth temperature to 10 °C and the lower optimum plant growth temperature to 30 °C. To expand the model’s use into areas with generally lower or higher background relative humidity the relation between weather and pan evaporation may have to be modified.

The model’s measurement of relative growth is most important to livestock producers. Across the landscape pastures differ in yield due the plant available water holding capacity of the soil [12], the growth potential of the plant species in the pasture, soil pH, soil phosphorus and potassium fertility, and nitrogen input. The purpose of this model is to estimate the relative effect of weather and soil water holding capacity on pasture growth determined by these agronomic management factors. Producers are working with pasture-livestock systems that they have developed with experience over time. A pasture growth model that describes the relative difference between the current year and an average year helps managers visualize the impact of the current weather relative to the average to see if management adjustments will be needed. When estimates of the effect of climate change on temperature and rainfall are available, these values can be used in the model to predict the impact of climate change on pasture growth within the region.

## 4. Materials and Methods

### 4.1. Pasture Growth Model Description

The pasture growth model, programed in Python, is provided in Appendix A. An example of the required weather data file is provided in Appendix B. The pasture growth model uses daily time steps to calculate the effects of air temperature and rainfall on pasture growth. Inputs for each day are day of the year, minimum temperature (t_min), maximum temperature (t_max), and precipitation (precip). Day of the year and latitude are used to predict potential solar radiation and evapotranspiration. Four factors that determine pasture growth are calculated: relative growth rate due to plant available soil water (ASW, rgr_asw), relative growth rate due to mean air temperature (rgr_temp), relative growth rate due to day length (rgr_dl), and relative growth rate due to soil biology (rgr_sbio). Relative growth rate due to the environment (rgr_env) is the product of these four values. Modelled daily rgr_env are summed over the growing season to calculate the cumulative relative growth rate due to the environment (cum_rgr_env).

The maximum plant available soil water (ASWmax) capacity for a soil is used to evaluate drought stress on plant growth. Identification and description of soils at a point in the USA landscape are available on the Soil Web [13] or Web Soil Survey [14]. Plant ASW capacity for a soil is listed under the soil’s hydraulic and erosion ratings.

Study year and long-term average weather data for the HOG site were obtained from NOAA [15]. The weather station used to monitor weather for the HOG site was 1.8 km from the pasture on a hill with the pasture being in the valley below. Study year weather data for the WVU site was measured using a recording weather station within the pasture, while long-term average weather history was obtained from a NOAA weather station located within the county.

### 4.2. Plant and Soil Biology Response to ASW

Evapotranspiration is based on a regional regression between weather station open pan evaporation and potential solar radiation, mean temperature, and an estimate cloud cover using the daily temperature range, (t_max minus t_min). This regression is based on observations in West Virginia [16,17] validated using observations from New York state (unpublished, NY monthly climatic summaries). Plant evapotranspiration as a fraction of open pan evaporation was obtained from research in the Northeast US [18,19]. Potential solar radiation is calculated based on latitude and day of year [20].

The model calculates ASW for a given day using the previous day’s ASW, adding the day’s precipitation and subtracting the day’s evapotranspiration. When this balance exceeds the soil’s ASWmax, ASW for the day is limited to ASWmax. Plant growth response to ASW (rgr_asw) is 1.0 when ASW is greater than 0.50 of ASWmax, decreasing to 0.0 as ASW goes to 0.0 of ASWmax [21,22]. This relation between ASW and ASWmax also applies to relative growth rate due to soil biology (rgr_sbio) that releases nitrogen and other nutrients from the soil organic matter to plants [21].

### 4.3. Plant Response to Temperature

Temperature impact on relative plant growth starts at 0.0 when air temperature is at a low minimum plant growth temperature (low_min_pgt), increases to 1.0 as temperature increases to a low optimum plant growth temperature (low_op_pgt), remains at 1.0 as temperature increases to a high optimum plant growth temperature (high_op_pgt), then decreases to 0.0 as temperature increases to a high maximum plant growth temperature (high_max_pgt). For cool-season (C3) plants these points are 0 °C, 10 °C, 20 °C, and 30 °C respectively [8].

### 4.4. Plant Response to Daylength

Photosynthetically active radiation effect on plant growth is estimated from daylength [8]. In the spring relative plant growth is 0.0 when daylength is less than 8 h, increases to 1.0 as daylength increases to 12 h. In the late summer relative plant growth remains at 1.0 until daylength decreases to 13 h then decreases to 0.0 as daylength decreases to 8 h. Daylength is calculated based on latitude and day of year [20].

The model initiates plant growth when annual summed daily temperature (Tsum) reaches 280 °C or 300 °C [23]. Sensitivity analysis found that early season growth was modelled best when growth starts at Tsum 280 °C at HOG and 300 °C at WVU.

### 4.5. Field Measurement of Pasture Growth

The pasture growth model was tested using observations from two pasture sites differing in grazing management, soil type, and elevation (Table 2).

One site was continuously stocked, mixed cool-season grass clover pasture, using variable stocking rate to manage height of grazing (HOG). At the HOG site, tester and grazer animals were used with the number of grazer animals being adjusted to maintain four ranges in sward heights: 4 to 6 cm, 6 to 8 cm, 8 to 10 cm, and 10 to 12 cm. This site had three replications of four HOG treatments for a total of 12 pastures studied over seven years providing 43 monthly comparisons of measured vs modelled forage growth. The HOG site is located in Monongalia County West Virginia, on the West Virginia University Experiment Station farm. The soil in these pastures is predominantly Culleoka series (Alfisol, fine-loamy, mixed, active, mesic Ultic Hapludalfs) which provides 9 inches of plant ASW.

The second site was rotationally stocked, mixed cool-season grass clover pasture where three rates of poultry litter were used to provide additional nitrogen (WVU). Four paddocks received poultry litter. Two paddocks received 4480 kg/ha in the spring. Another two paddocks received 4480 kg/ha in spring and an additional 4480 kg/ha in the fall. Four paddocks received no poultry litter. Pasture swards were grazed when sward ruler height averaged 27 cm to a residual height of 12 cm. This provided linear, linear-plateau, and exponential forage growth 50, 42, and 8- percent of the time, respectively under good growing conditions [25]. This site had two replications of four paddock each for a total of 12 pastures studied over three years, providing 79 grazing events to compare measured vs modelled forage growth. The WVU site is located in Preston County West Virginia, on the West Virginia University Experiment Station Farm, Reedsville. The soil in these pastures is predominantly a Gilpin series (Ultisol, fine-loamy, mixed, active, mesic Typic Hapludults). This soil provides 5-inches of ASW.The predominant forages at the two sites were Tall fescue (*Schedonorus arundinaceus* previously known as *Festuca arundinacea*, Schreb.), orchardgrass (*Dactylis glomerata* L.), timothy (*Phleum pratense* L.), Kentucky bluegrass (*Poa pratensis* L.), red top bentgrass (*Agrostis alba* L., previously known as *Agrostis gigantia*), red clover (*Trifolium pratense* L.), white clover (*Trifolium repens* L.) and mixed forbs.

Mean monthly temperatures during the study were near the long term normal (Figure 3a,b). Total monthly rainfall was variable about the normal with below normal rainfall occurred in 1993 and 1999 (Figure 3c,d). The HOG site is warmer and slightly drier than the WVU site due to being at a lower elevation.

### 4.6. Forage Growth Measurement

Forage growth was measured at the HOG site by using exclusion cages. Exclusion cages were set at random within treatment paddocks and forage allowed to grow for 12 to 35 days unless growth was exceptionally fast or slow in which case forage growth was measured over shorter or longer time periods. Forage height within the exclusion cage was measured using a standard resting plate meter [26]. Forage mass was estimated using plate meter calibration based on paired plate meter height and forage samples clipped at soil surface obtained over the growing season. Forage height and mass were measured the same way at the end of growth intervals. Average forage growth rate (kg DM/ha/day) during the growth period was calculated. Growth rates were assigned to midpoint day of the year within the growth interval. Growth rates were averaged by month and cumulative growth at the end of each month were summed for the year. Relative growth rates were calculated using the highest cumulative growth rate observed across years (9119 kg/ha).

At the WVU site forage height was measured for each grazing event at 30 points within pastures before and again after grazing. A standard resting plate meter [26] was used to measure forage height. At 15 of the 30 forage height sample points paired forage mass samples were clipped at the soil surface. The paired forage mass to resting plate meter height samples were used to calibrate forage plate meter height to forage mass pre-grazing and post-grazing. The amount of forage grazed off the pasture was considered the effective forage growth [27,28]. Forage growth grazed from each paddock was summed over the year to give cumulative forage growth for the paddock. Relative growth rates (relFGobs) were calculated using the highest cumulative growth rate observed across years (7529 kg/ha).

### 4.7. Testing the Model

To test the pasture growth model’s ability to describe observed pasture growth the independent variable cum_rgr_env was regressed against the dependent variable cumFGobs [29]. The test criteria were regression R^2^, the standard deviation of residual values about the regression (SDreg, also known as the square root of mean square error), and average absolute percent error (AAPE). High R^2^ and low SDreg and AAPR indicate high descriptive ability of a model. Year and treatment effects were tested for significance (*p* = 0.05) and accounted for when present.

## Figures and Tables

**Figure 1 plants-10-01754-f001:**
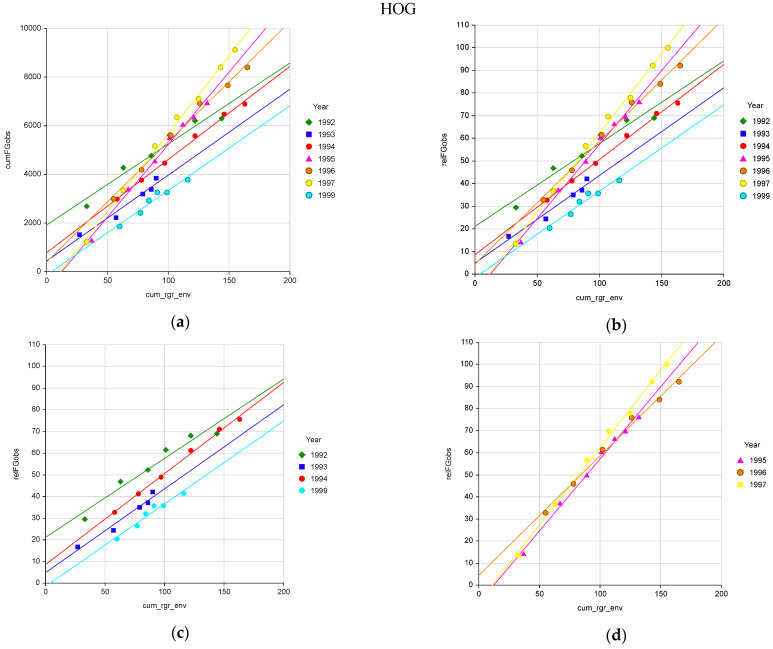
(**a**) Accuracy of the pasture growth model’s cumulative relative growth rate due to environment (cum_rgr_env) describing cumulative forage growth observed (cumFGobs, kg/ha) and (**b**) cumulative relative forage growth observed (relFGobs) at the HOG location over seven years with (**c**) years showing individual year effects (different intercepts) and (**d**) years showing a consistent proportional effect (different slope).

**Figure 2 plants-10-01754-f002:**
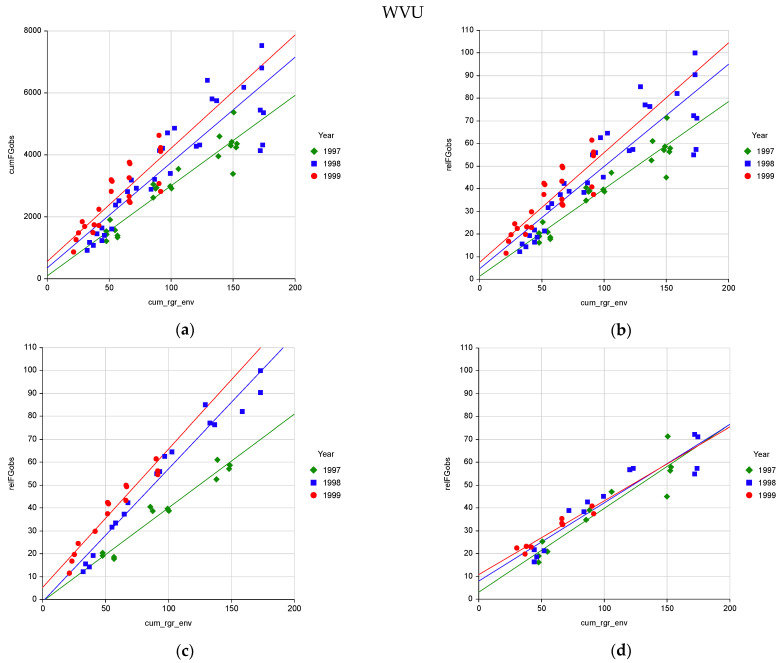
(**a**) Accuracy of the pasture growth model’s cumulative relative growth rate due to environment (cum_rgr_env) describing cumulative forage growth observed (cumFGobs) and (**b**) cumulative relative forage growth observed (relFGobs) at the WVU location over three years with (**c**) poultry litter treatments and (**d**) no-litter treatments showing the effect of cumulative nitrogen from poultry litter increasing pasture growth over time.

**Figure 3 plants-10-01754-f003:**
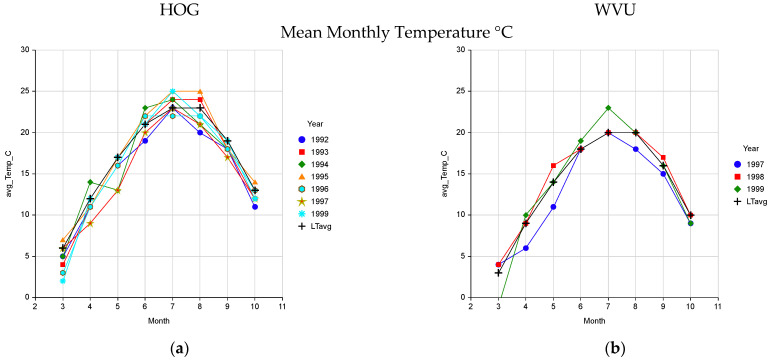
Mean monthly air temperature at the HOG (**a**) and WVU (**b**) sites and total monthly rainfall and long tern averages (LTavg) for these weather components at the HOG (**c**) and WVU (**d**) sites used to validate the pasture growth model.

**Table 1 plants-10-01754-t001:** Accuracy of the pasture growth model’s cumulative relative growth rate due to the environment (cum_rgr_env) for describing cumulative forage growth observed (cumFGobs, kg/ha) and relative forage growth observed (relFGobs ^†^) at two sites over multiple years.

Regression		R^2^	SD_reg_ ^‡^	AAPE ^‡‡^	N
HOG					
A	cumFGobs = −232 + 51 cum_rgr_env	0.83	849	18	43
B	relFGobs = −3 ^††^ + 0.56 cum_rgr_env	0.83	9	18	43
C	For years 1992, 1993, 1994, 1999 relFGobs = 19 + 0.39 cum_rgr_env + Year Year: 1992 = 0 1993 = −14 1994 = −7 1999 = −21	0.98	2	4	23
D	For years 1995, 1996, 1997 relFGobs = −7 + 0.64 cum_rgr_env + Year Year: 1995 = 0 1996 = 0 1997 = 5	0.98	3	7	20
WVU					
E	cumFGobs = 578 + 30 cum_rgr_env	0.76	759	22	79
F	relFGobs = 8 + 0.40 cum_rgr_env	0.76	10	22	79
G	relFGobs = -9 + 0.44 cum_rgr_env + 9 Litter + Year Year: 1997 = 0 1998 = 9 1999 = 14	0.87	7	17	79
H	No-Litter Treatments relFGobs = 8 + 0.34 cum_rgr_env	0.88	6	14	35
I	Litter Treatments in 1997 relFGobs = −1 ^††^ + 0.41 cum_rgr_env	0.95	4	9	12
J	Litter Treatments in 1998 and 1999 relFGobs = −2 ^††^ + 0.60 cum_rgr_env + Year Year: 1998 = 0 1999 = 9	0.96	6	11	17

^†^ Maximum cumFGobs was 9,119 and 7,529 kg/ha at the HOG and WVU sites, respectively. ^††^ Not significantly different from zero at *p* = 0.05. ^‡^ SDreg standard deviation of residuals about the regression. ^‡‡^ AAPE average absolute percent error.

**Table 2 plants-10-01754-t002:** Environment and management of sites where pasture growth was measured.

Site	MLRA ^†^	Elevation	Grazing Management	Treatments	Years	Growth Intervals
HOG	126	305 m	Set stocked	Height of grazing	7	43
WVU	127	610 m	Rotational stocked	Poultry litter applications	3	79

^†^ Major Land Resource Areas, USDA/NRCS [24]. 126—Central Allegheny Plateau. 127—Eastern Allegheny Plateau and Mountains.

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
