# Peer review of "Validating a Simple Mechanistic Model That Describes Weather Impact on Pasture Growth"

_plants, 2021, doi:10.3390/plants10091754_

Round 1
Reviewer 1 Report
This manuscript presents a pasture growth model to predict growth by weather variables. It is an interesting approach and can be a useful tool in agricultural management planning. The manuscript is quite short, and in my opinion, the author could give more background information on the potential usability of such a model, in particular in course of climate change. In the discussion more information on management implications should be given. I also do not really get the issue with the year effect. As you demonstrate in line 72 to 78, the model captures the differences in weather conditions (mainly water availability) between the years and as I understood your approach, this is the main aim of the approach, to be able to predict growth when future weather conditions may be modelled. Thus, could you make more clear what you mean with “There was a year effect that the model did not describe…”. I also think that describing the regression lines in Figure 1 and 2 in more detail would be good. It becomes not exactly clear from reading the text that the lines showing no year effect can be interpreted as an average baseline growth with drier years (those lines with a year effect) showing an increasing reduction from the baseline with growth accumulation. In Figure 1 c, d the colours and symbols for the years should be the same as in Figure 1 a, b. L66: its Fig. 1Author Response
Reviewer 1
This manuscript presents a pasture growth model to predict growth by weather variables. It is an interesting approach and can be a useful tool in agricultural management planning. The manuscript is quite short, and in my opinion, the author could give more background information on the potential usability of such a model, in particular in course of climate change.
I added to the text to address this comment.
In the discussion more information on management implications should be given.
I added to the text to address this comment.
I also do not really get the issue with the year effect. As you demonstrate in line 72 to 78, the model captures the differences in weather conditions (mainly water availability) between the years and as I understood your approach, this is the main aim of the approach, to be able to predict growth when future weather conditions may be modelled. Thus, could you make more clear what you mean with “There was a year effect that the model did not describe…”.
I added to the text the fact that the weather station used for the HOG site was not located at the field so the weather impacting the growth of forage was slightly different than the weather recorded at the weather station. At the WVU site the weather was measured at the edge of the pastures.
I also think that describing the regression lines in Figure 1 and 2 in more detail would be good. It becomes not exactly clear from reading the text that the lines showing no year effect can be interpreted as an average baseline growth with drier years (those lines with a year effect) showing an increasing reduction from the baseline with growth accumulation.
The answer to this comment is in Table 1 where the statistical impact of year is described.
In Figure 1 c, d the colours and symbols for the years should be the same as in Figure 1 a, b. L66: its Fig. 1
I made this correction.
Reviewer 2 Report
To measure the relative growth is very important to livestock producers, it is very good to develop a pasture growth model which requires only latitude and daily maximum and minimum temperature and rainfall. It based on a large number of measured data and analysis, the simulation results are also very good. The result would be very nice for the livestock producer to predict the impact of climate change on pasture growth. However, the core content of the model is not explained clearly in this manuscript. There are some comments and suggestions:
- The main conclusions are not clear. What is the impact of climate change on pasture growth? There is no clear statement.
- It is good to have the python code of the model. But still it would be better to have some introduction of the model. What is the structure and the main composition of the model? It makes me take a long time to read the code of the model. If there is a description, it would be much easier.
- How did the author revise the parameters to make the simulation results of the model more perfect?
- Can this model be extended to other areas, or is it only applicable to this study area? Does the change of vegetation type affect the model?
- Line 11 “at a site under continuous stocking (HOG) and a site under 11 rotational stocking (WVU).” When HOG and WVU come at the first time, it makes me confused.
- In Figure 1, The color of the legend of (b)(c)(d) should be consistent,it would be more convenient for reading and comparative analysis
Author Response
Reviewer 2
To measure the relative growth is very important to livestock producers, it is very good to develop a pasture growth model which requires only latitude and daily maximum and minimum temperature and rainfall. It based on a large number of measured data and analysis, the simulation results are also very good. The result would be very nice for the livestock producer to predict the impact of climate change on pasture growth. However, the core content of the model is not explained clearly in this manuscript. There are some comments and suggestions:
- The main conclusions are not clear. What is the impact of climate change on pasture growth? There is no clear statement.
-
- Correct, there is no clear statement since the impact of climate change differs from location to location and there is not one answer that applies across the landscape. In some places it is predicted that rainfall will increase while in other areas rainfall may decrease. In general temperatures are expected to increase but at different rates depending on the location. That is why the concluding sentence is given as it is.
-
- It is good to have the python code of the model. But still it would be better to have some introduction of the model. What is the structure and the main composition of the model? This is already addressed in the original text. It makes me take a long time to read the code of the model. If there is a description, it would be much easier. .
- I did not make this addition since I do not see an easier way of presenting the model then what is provided in the text and the Python code.
- How did the author revise the parameters to make the simulation results of the model more perfect? This paper does not address making improvements to the model to make it “more perfect”. The purpose of the paper is to validate the use of integrating previously reported research-based algorithms for predicting pasture growth due to weather.
- I agree there is value to modifying the model but then the modified model will need to be validated against another set of independently observed pasture growth to validate the modifications. That is beyond the purpose and scope of this paper.
- Can this model be extended to other areas, or is it only applicable to this study area? Does the change of vegetation type affect the model?
- I added to the text to address this question.
- Line 11 “at a site under continuous stocking (HOG) and a site under 11 rotational stocking (WVU).” When HOG and WVU come at the first time, it makes me confused.
- I added to the text to address this comment.
- In Figure 1, The color of the legend of (b)(c)(d) should be consistent,it would be more convenient for reading and comparative analysis
- I made this correction.
Reviewer 3 Report
I liked this article because is pretty straightforward. I have one major concern in regard of the Introduction.
M&M, Results and discussion are well detailed but the Introduction is too superficial. I'd recommend rewriting the Intro with more citations and a better description of the problem. Right now is too much on the math side but not much on the plant side.
Author Response
Reviewer 3
I liked this article because is pretty straightforward. I have one major concern in regard of the Introduction.
M&M, Results and discussion are well detailed but the Introduction is too superficial. I'd recommend rewriting the Intro with more citations and a better description of the problem.
I added to the text to address this comment.
Right now is too much on the math side but not much on the plant side.
This paper is about a mathematical plant growth model. The equations are mathematical descriptions of how plants respond to daylength, air temperature and available soil water. The intent is not to discuss the plants physiological responses during photosynthesis, or to temperature and drought stress. The math is the essence of the model, the validation of the model, and there for the manuscript.
Round 2
Reviewer 3 Report
The Author addressed all my comments and the manuscript is now ready for publication.